# Carbon-Based Materials and Their Applications in Sensing by Electrochemical Voltammetry

Trong Danh Nguyen [†], My Thi Ngoc Nguyen [†] and Jun Seop Lee *

Department of Materials Science and Engineering, Gachon University, 1342 Seongnam-Daero, Sujeong-Gu, Seongnam-Si 13120, Gyeonggi-Do, Republic of Korea
* Correspondence: junseop@gachon.ac.kr; Tel.: +82-31-750-5814
† These authors contributed equally to this work.

**Abstract:** In recent years, society has paid great attention to health care and environmental safety. Thus, research on advanced sensors for detecting substances that can harm health and the environment has been developed rapidly. Another popular target for detection techniques is disease-expressing materials that can be collected from body fluids. Carbon, which has outstanding electrochemical properties, can come from a variety of sources and has many morphological shapes, is nevertheless an environmentally friendly material. While carbon nanomaterial has become one of the most common targets for high-tech development, electrochemical voltammetry has proven to be an effective measurement method. Herein, the paper proposes a currently developed carbon nanomaterial along with research on a modified carbon material. Moreover, four common voltammetry methods and related works are also introduced.

**Keywords:** carbon materials; electrochemical; voltammetry; detection





## 1. Introduction

Electrochemistry is a science that studies the relationship between electricity where the applied potential and the output current are recorded continuously. A suitable applied potential can trigger an available redox reaction, and this reaction can also affect the record current. While the oxidation reaction can release additional energy, the reduction reaction will consume energy, making a change in the current flow. It is possible to detect such changes in the current, thus recognizing whether the redox reaction in the electrolyte exists. Hence, electrochemical methods are a powerful tool for detecting substances in the electrolyte such as biomaterials or ions [1,2]. The need for such detection has attracted great attention because the presence of these substances can have a huge impact on the environment and human health [3,4]. Voltammetry methods, on the other hand, rely on applying different voltages to find the change in current at which the reaction occurs. For the purposes of electrochemical detection, many voltammetry measurement methods have been developed, including linear sweep voltammetry (LSV), cyclic voltammetry (CV), different pulses voltammetry (DPV) and square wave voltammetry (SWV) [5,6]. Until recently, other methods were developed to optimize the sensitivity as well as measurement ranges [7].

Carbon-based material is a low cost, environmentally friendly material. Though carbon is one of the most basic nanomaterials, it still attracts a lot of attention due to its outstanding properties in terms of electrical conductivity, fast charge transfer, high stability and ease of modification [8–10]. Attempts were made not only to alter the morphological structure of carbon but also to modify the elements contained in its chemical structure [11]. While the morphology of the material can easily be altered for better surface area and large pore sizes, the result is usually an amorphous carbon material. The chemical structure of carbon can be modified by both physical and chemical methods [12,13]. It has been proven that functional groups can significantly improve the electrochemical properties of

the carbon material [14]. Up until now, there have been various studies on the modifications of the carbon material.

Because carbon and its variations possess excellent electrochemical properties, they have been used in electrochemical applications [15]. Without exception, carbon nanomaterials have also been used in voltammetry measurement methods [16–18]. The materials have been developed so that they can detect the smallest amount of target while maintaining the linearity of the peak current (linear range) [19]. The measurement targets are usually heavy metal ions or biomaterials which can cause diseases or pollute the environment [20]. The other well-known targets are the chemicals that express the existence of cancer cells, stress, disease, viruses and anti-cancer drugs [21–25]. The collected data can be used to predict the disease and mental state of patients.

Current development in carbon materials and their application in the field of voltammetry are omitted from this review manuscript. First, the types of carbon were classified and compared according to their outstanding properties. Among the materials that should be mentioned are graphene oxide (GO)/graphene oxide (rGO), amorphous carbon with different morphologies, biomass-derived carbon (natural source carbon) and metal/metal oxide–carbon composite. Subsequently, applications of the carbon material in detecting targets by voltammetry were proposed. There, the currently commonly used voltammetry methods are discussed and the advantages of voltammetry in detection are proposed.

## 2. Sp$^2$ Carbon Materials

As mentioned in the Introduction, the hydrophilicity–hydrophobicity and electrochemical properties of carbon materials can be modified by incorporating different types of chemical functionalities and heteroatoms (O, N, B, S or P) on the carbon surface by physical and chemical methods [26]. Besides, biomass is one of the abundant natural carbon sources, which can be self-doped and made during synthesis due to the available structural components (S, N and P elements) [27]. Therefore, many carbon-based material structures have been proposed. This review will focus on the morphology, advantages and disadvantages of producing graphene oxide/reduced graphene oxide, amorphous carbon, biomass-derived carbon and metal/metal oxide–carbon composites which have great potential in the application of electrochemical voltammetry (Figure 1).

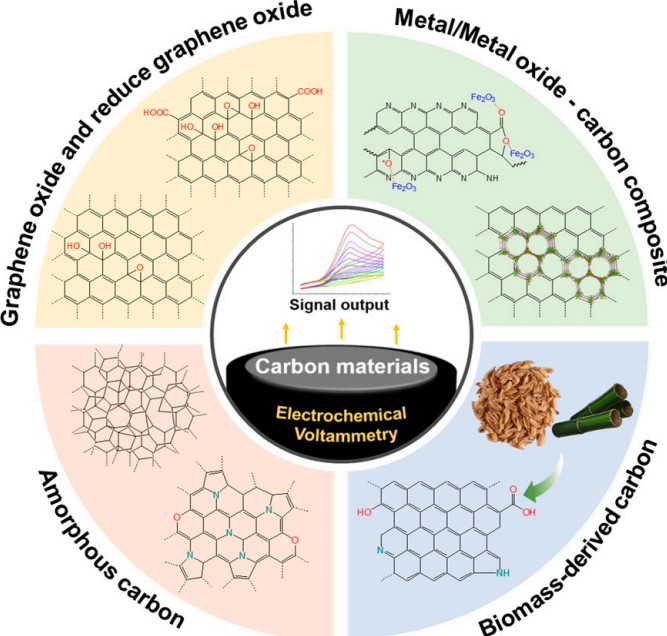

**Figure 1.** Carbon materials are commonly used to construct electrochemical voltammetry electrodes.

## 2.1. Graphene Oxide and Reduced Graphene Oxide

Thanks to its unique optical, mechanical, chemical and electronic properties, GO has become one of the most promising materials that has been extensively studied in the past decade [28]. So far, many studies have been conducted using GO related to energy storage, solar cells, desalination, oil–water separation, drug delivery, etc. [29–33]. GO is structurally similar to graphite, though the base plane and edges of GO contain hydroxyl, carboxyl and epoxy oxygen functional groups, leading to $sp^2$ and $sp^3$ hybridization of the carbon in the structure. Because GO has good solubility in water/organic solvents and has a favorable electron mobility structure, it becomes an attractive material for electrochemical studies and applications. Normally, graphene oxide can be generated by treating graphite in a strong oxidizing environment, including Brodie's method, Hummers' method, Tour's method and the electrochemical method, which will be described in more detail below [34,35]. Hummers' method is commonly used to synthesize GO, but this method is not environmentally friendly and has a low yield [36]. Therefore, Marcano et al. propose Tour's method (improved graphene oxide/IGO) which can overcome the disadvantages of previous ones. Compared to Hummers' method, this method uses more $KMnO_4$, and the reaction takes place in the presence of a mixture of $H_2SO_4$ and $H_3PO_4$ (9:1 ratio) [37] (Figure 2a). Here, the presence of $H_3PO_4$ increases the efficiency of the oxidation process [38]. In order to demonstrate the superiority of the IGO method over Hummers' method (HGO) and modified Hummers' method ($HGO^+$), the author compared the volume of under-oxidized materials recovered after the process. As a result, for the same amount of starting material, IGO produced significantly less amount of hydrophobic carbon material than either HGO or HGO. In addition, the synthesis process does not emit a large amount of heat and toxic gases, which is an advantage when producing GO on an industrial scale. With the trend towards finding an environmentally friendly GO synthesis method, Pei et al. synthesized pure GO sheets by the water electrolytic oxidation of graphite [39] (Figure 2b). In this study, commercial flexible graphite paper was used as the raw material for the synthesis of GO in two sequential electrochemical processes taking place at room temperature. Initially, the sliced graphite paper undergoes electrochemical intercalation in a concentrated $H_2SO_4$ solution, which causes a significant decrease in surface resistance. In the second electrochemical process, the acidic solution is diluted, and the anodic electrocatalytic oxygen evolution of water occurs under the applied voltage. Compared to Hummers' and $K_2FeO_4$ methods, this method is about 100 times faster in oxidation, requires less $H_2SO_4$ and can be fully reused.

The negatively charged functional groups in GO facilitate the formation of stable colloids in water and are an ideal candidate for some applications such as sensors, super-capacitors, multifunctional gels, etc. [40–42]. However, in some cases, these functional groups also become a disadvantage as they create defects in the GO crystal lattice and make GO susceptible to degradation upon hydration. To overcome the disadvantages of GO, researchers performed GO reduction using the electrochemical, microwave, photo assisted and thermal methods [43–46] to obtain reduced GO (Figure 3).

However, since the reduced amount of functional groups leads to agglomeration and a tendency for reduced GO to crumble, the synthesis of reduced-GO membranes is still challenging. Recognizing the need for further studies on the effect of oxygen-containing functional groups on the success of reduced-GO-membrane synthesis, Huang et al. conducted a study to clarify this issue [47] (Figure 2c). In this study, a hydrothermal method was used to synthesize a uniformed reduced-GO membrane. Here, the hydroxyl and carboxyl groups are mainly responsible for the formation of a homogenous reduced-GO film. Due to hydrogen bonding between adjacent layers formed by hydroxyl and carboxyl functional groups, the surface interaction between the reduced-GO sheets is enhanced. Then, a reduced-GO film was synthesized without shrinkage or damage even after drying. Moreover, this study is the premise for several future studies related to the hydrothermal synthesis of rGO, which can be applied in enhancing photocatalysis and improving the electrochemical performance of electrodes.

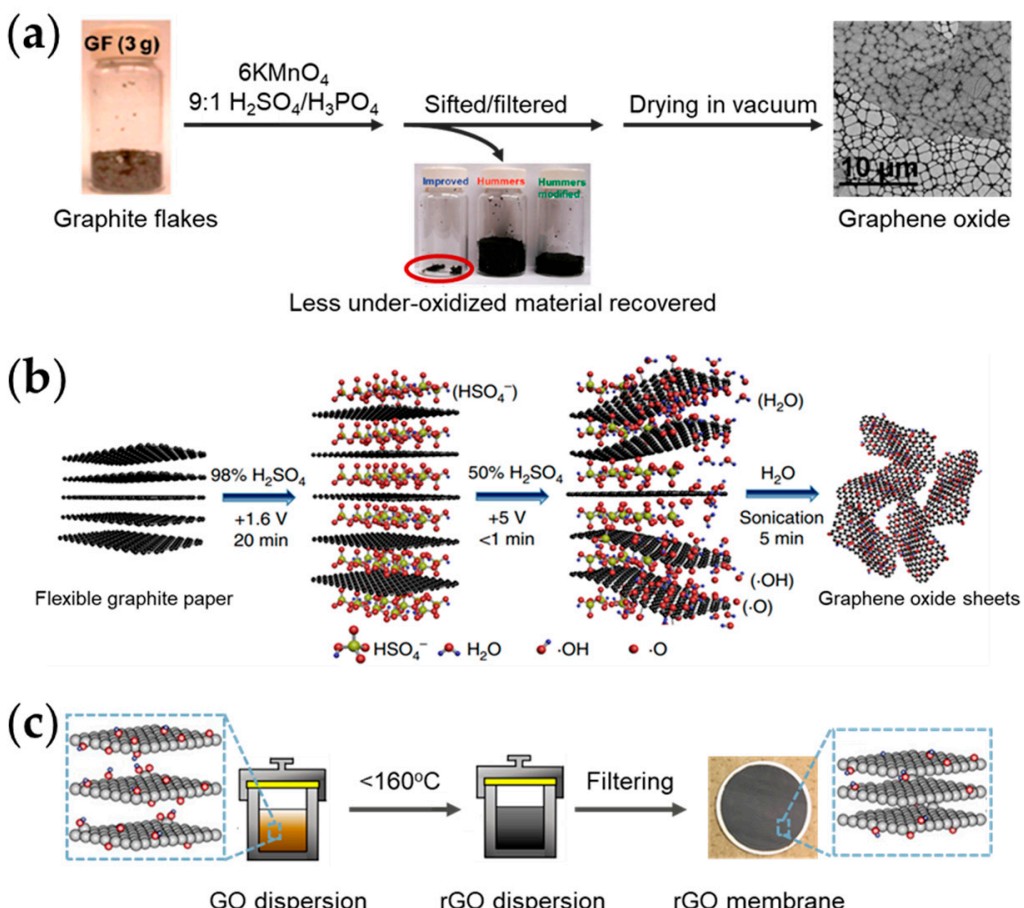

**Figure 2.** (**a**) Synthesis of graphene oxide by the improved method gives a higher yield of the product than the traditional method [37]. (**b**) GO synthesis by two sequential electrochemical processes with ultrafast reaction times [39]. (**c**) Synthesis of a homogeneous reduced GO membrane by hydrothermal method [47].

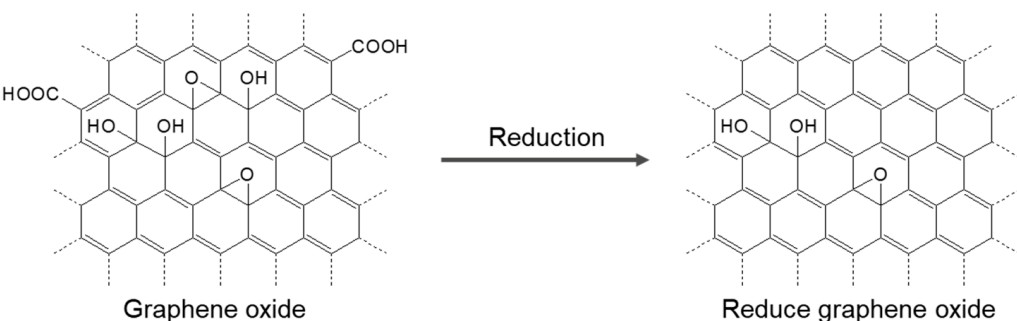

**Figure 3.** Chemical structure of GO and reduced GO produced after reduction.

## 2.2. Amorphous Carbon

Amorphous carbon is one of many allotropic forms of carbon, including carbon materials that have no long-range crystalline order in their structure [48] (Figure 4). Amorphous carbon includes $sp^3$ (can give a three-dimensional structure) and $sp^2$-hybridized carbon atoms (can form chains, benzene-like rings, hexagon-based layers, etc.), leading to disorder and multidimensionality in the structure [49].

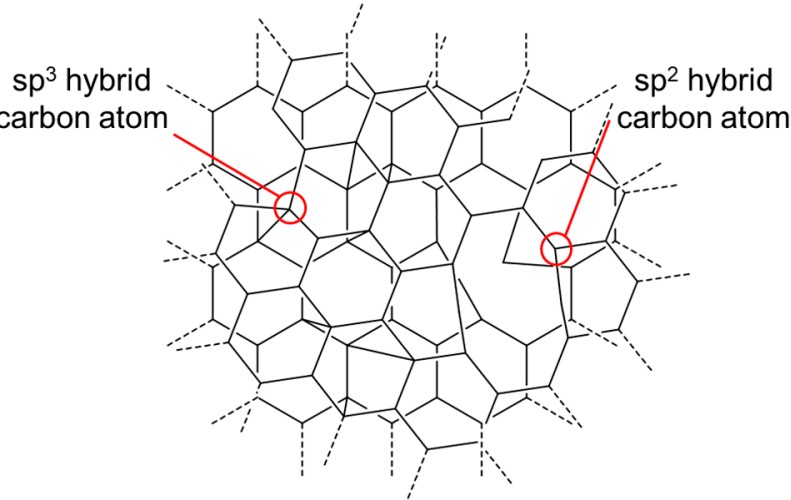

**Figure 4.** Chemical structure of amorphous carbon.

As a result of incomplete pyrolysis, amorphous carbon substances of plant and animal origin can be produced. Moreover, they can also be made of polymers such as polyacrylonitrile, polyimide, polyvinyl alcohol, etc. [50–52]. Through the carbonization process at a suitable temperature range, a short-range order carbon structure (amorphous carbon) will be created (Figure 5).

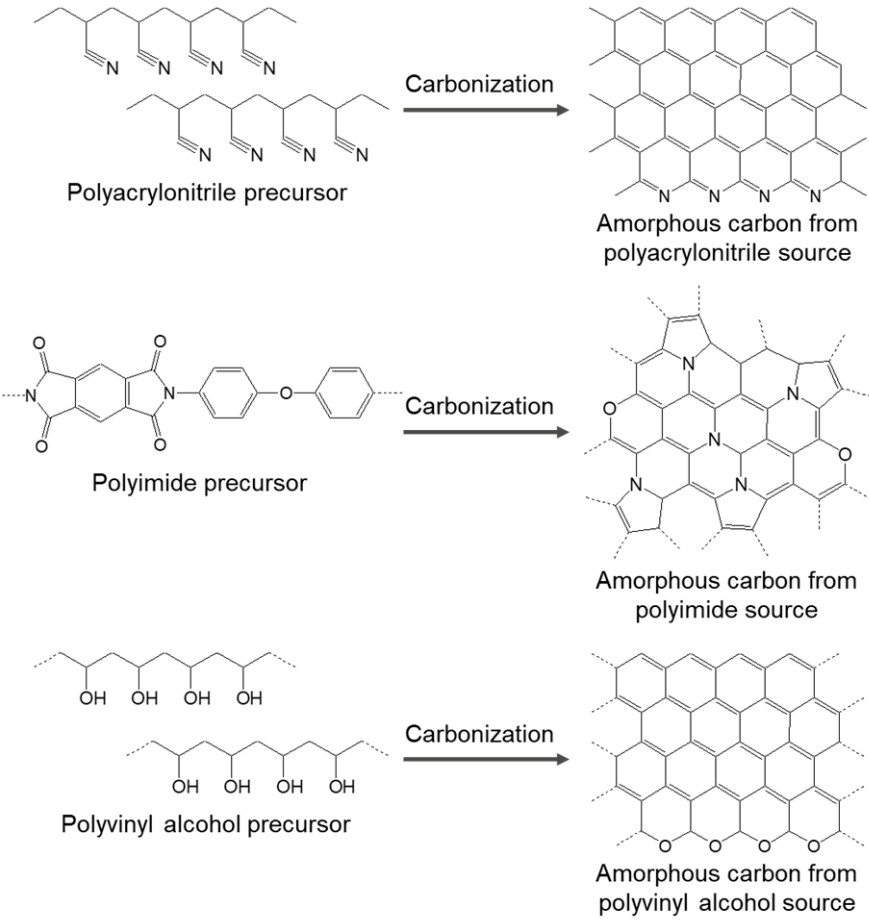

**Figure 5.** Chemical structures of amorphous carbon after carbonization polyacrylonitrile, polyimide, polyvinyl alcohol precursor, respectively.

Kim et al. synthesized activated multi-size pores containing carbon nanofibers using single nozzle co-electrochemistry and a steam-activated process to generate a signal transducer and template for immobilization bioreceptors [53] (Figure 6a). Here, the electrospinning method is used to synthesize a 1D structure (fiber). This method allows control over the structure of the material by varying the composition of the precursor. For instance, polyacrylonitrile can undergo carbonization in the temperature range of 800–1300 °C, and poly methyl methacrylate will be degraded at about 300 °C. Therefore, poly acrylonitrile acts as the carbon source of material, and the decomposition of poly methyl methacrylate creates the pore structure for the carbon nanofibers. The steam activation step maximizes the amount of micro/mesoporous, resulting in an increased surface area of the material, meaning that the antibody can easily coat the sensing electrode and increase its affinity for the target analyte.

Realizing that the fully $sp^3$-bonded amorphous carbon structure has the potential to achieve the unique properties of both diamond crystals and amorphous materials, Zhi-dan Zeng et al. synthesized it by combining high-pressure and in situ laser heating techniques [54] (Figure 6b). Glass carbon with $sp^2$ bonds is the precursor to this process. After loading the glassy carbon sample with high pressure > 40 GPa and a temperature of about 1800 K, carbon with a complete $sp^3$ bond was formed with a disordered structure (amorphous diamond). X-ray diffraction experiments revealed that amorphous carbon has good incompressibility comparable to crystalline diamond and generally has much better properties than glassy carbon. The discovery of the carbon material containing a fully $sp^3$ amorphous structure with outstanding physical properties added a diamond-like structure to the carbon family. Such material seems to have a high potential for further exploitation.

Amorphous carbon thin film is also an allotropic form of carbon, which is of great interest to scientists due to its outstanding mechanical and electrochemical properties and its outstanding potential for applications in arrays systems such as solar panels, hard masks, deformation and connection electrodes, sensors, micro-supercapacitors, batteries, nanogenerators, etc. [55]. Specifically, in a study involving foldable electronics, Pal et al. synthesize hydrogenated amorphous carbon thin films and use them as ultra-thin anisotropic conductive films to coat a printed metal circuit board [56] (Figure 6c). Branched poly(ethylenimine) is the starting material, acting as a carbon source and nitrogen dopant. The dimethylformamide solvent was used to dissolve the polymer to form a precursor solution. The precursor solution was then immobilized on a conductive electrode plate (Si wafer or ITO) by the spin-coating method. The sample was decomposed in a microwave oven with a power of 240 W (to set the temperature for the substrate at 350 °C) for 1 min to produce a thin film of nitrogen-doped amorphous carbon. It is a fast, one-step carbon film synthesis process that shows great potential for practical manufacturing applications.

### 2.3. Biomass-Derived Carbon

As we all know, biomass is the combination of carbon with numerous other elements (H, O, N, P, S, etc.) to create living biological organisms in nature. Heteroatom-doped three-dimensional carbon materials with a porous structure and made of biomass have great potential for electrochemical applications [57–59].

Bamboo is well known as an inexpensive biological source due to its short maturation cycle and ability to grow almost anywhere [60]. Furthermore, it is rich in silicon dioxide (precursor of silicon carbide (SiC)) and nitrogen-containing functional groups (precursor of pyrrolic-N), which can contribute to improving the electrochemical properties of carbon materials and the overall performance of supercapacitors [61]. Realizing the promising potential of bamboo, Abbas et al. used it as a starting material for the synthesis of hierarchically porous bio-renewable carbon materials doped with SiC/pyrrolic-N [62] (Figure 7a). After the pyrolysis process, the nitrogen-containing functional groups in natural bamboo will produce pyrrolic-N species, resulting in increased wettability and better electrolyte ions transfer through the carbon material. The carbonized powder was activated with potassium

hydroxide. In this process, $SiO_2$ becomes a sacrificial material, leaving an ultra-microporous and mesoporous surface of the carbon material.

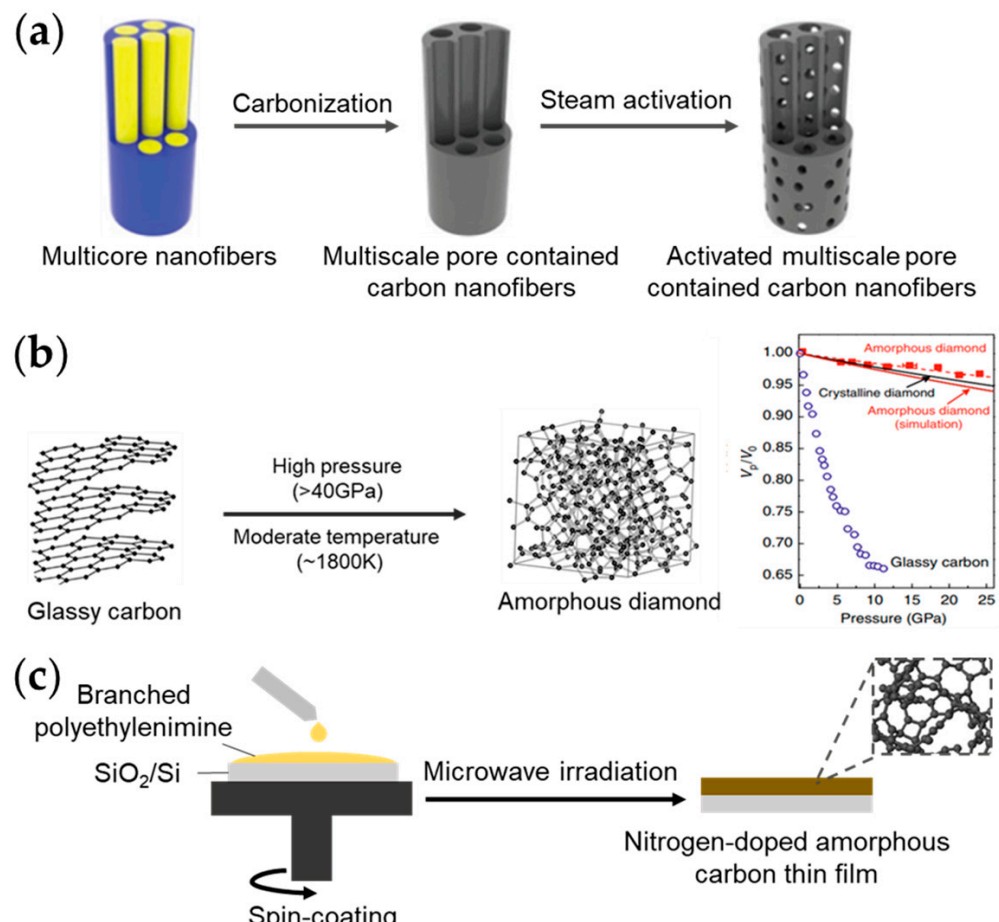

**Figure 6.** (**a**) High-area carbon nanofibers are synthesized by electrospinning and heat treatment [53]. (**b**) Structure of fully $sp^3$-bonded amorphous carbon synthesized from vitreous carbon by in situ and high-pressure laser heating technique [54]. (**c**) Thin films of hydrogenated amorphous carbon are synthesized by spin-coating and microwave irradiation [56].

In addition to the direct use of sources from living organisms, using by-products from biomass processing for electrochemical applications is one of the ways to reduce input material costs in device production. Heavy bio-oil is an industrial by-product, characterized by high viscosity and poor fluidity, which makes its use very limited [63]. Thanks to the composition and elemental analysis, Zhu et al. found that the C and O content of heavy bio-oil was 57.36% and 36.05%, respectively, showing that it is an ideal carbon source with oxygen-containing functional groups. To open a new direction for the reuse of this waste source, Zhu et al. proposed the preparation of hierarchical porous carbon material from heavy bio-oil derived from the pyrolysis of rice husk [64] (Figure 7b). The carbonization of heavy bio-oil under nitrogen atmosphere is the first stage of the synthesis process. The carbonation product undergoes activation, where the activating agent is sodium hydroxide. Upon completion of both processes, a three-dimensional bonded porous structure with surface oxygen-containing functional groups is obtained. The resulting material applied to the supercapacitor exhibits excellent capacitance performance exceeding expectations in comparison with the carbon material produced directly from the raw husk [65].

The conversion of biomass to carbon materials for stable solid carbon storage is a solution to reduce global emissions known as negative emission technology [66]. The hydrothermal method only uses high-pressure water as a solvent to convert biomass into carbon materials [67]. Although easy to operate, pressure and temperature in batch reactors vary proportionally, making it difficult to determine which factor affects the final product [68,69]. To overcome that shortcoming, Yu et al. developed a process to independently control temperature and pressure in a hydrothermal system that can heat cellulose to form submicron carbon spheres [70] (Figure 7c). In the presence of water, under constant pressure (20 MPa) and with a gradual increase in temperature, the cellulose structure decomposed from flat, rod-shaped cellulose to crystalline cellulose (<100 °C) and finally to a carbon sub-micron sphere (100–150 °C). In this process, water at high pressure plays a key role as it has the ability to cut hydrogen bonds, leading to an increase in the rate of cellulose decomposition at a low temperature of 117 °C instead of above 200 °C as in previous studies. Compared to the conventional cellulose hydrothermal reaction, this hydrothermal system not only has a fast reaction time and low temperature, but the size of the sub-micron carbon spheres produced is about 40 times smaller. This study has contributed to a new approach to the production of sustainable carbon materials.

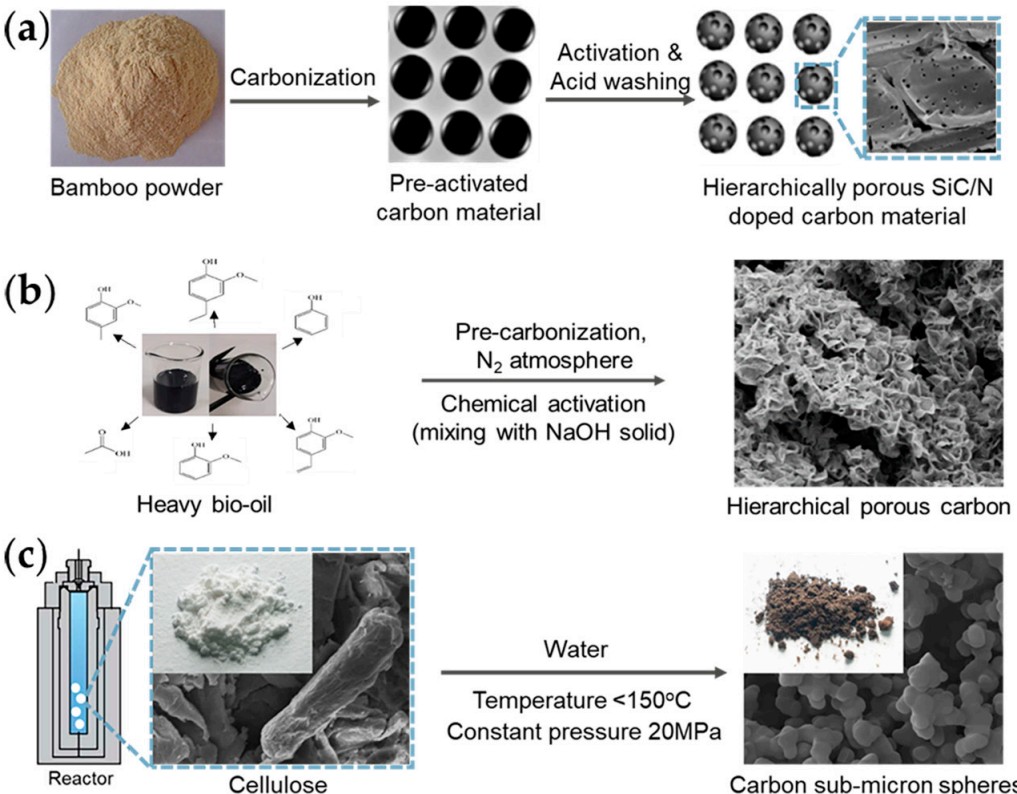

**Figure 7.** (**a**) A carbon material with a dual-doped SiC/pyrrolic-N pore structure is generated from the pyrolysis and activation of bamboo powder [62]. (**b**) After heat and activation treatment, the rice husk pyrolysis by-product is converted into a three-dimensional bonded porous structure with surface oxygen-containing functional groups [64]. (**c**) Synthesis of submicron carbon spheres from cellulose by the hydrothermal system (independently controlling temperature and pressure) [70].

## 3. Metal/Metal Oxide–Carbon Composite Materials

Metal/metal oxide–carbon composite materials are emerging due to their fascinating physical, chemical, electrical, and optical properties, as well as a wide range of applications (sensors, supercapacitors, automobiles, etc.) [71]. Unlike the material that was mainly built based on the carbon structure, the carbon composite was designed so that the metal/metal oxide has the largest active surface area. The composite's detecting function benefits from

the loaded material. Here, $sp^2$ carbon plays the role of the bone structure, which conducts the electricity between the electrode and the loaded materials. Instead, the metal/metal oxide material exchanges electrons with substances in the electrolyte. Though the metal oxide material does not high electrical conductivity like metal materials, they both have excellent electrocatalyst ability, resulting in higher sensitivity [72,73]. It is noted that some of the metal oxides like $SnO_2$ and $ZnO$ can carry the properties of a semiconductor material, which made them good materials in electron transfer ability [74,75]. $Fe_3O_4$ and $Co_3O_4$ with multiple valence states have also been found to be outstanding in their performance in catalytic activities [76,77].

A method for dispersing gold nanoparticles in carbon microspheres that is non-toxic has been proposed by Zhou et al. (Figure 8a) [78]. The $Au^{3+}$-decorated microspheres were obtained by simply dissolving $HAuCl_4$ in the suspension of polydopamine. Here, polydopamine acts not only as an adsorbent, reducing agent and stabilizer but also as a carbon source for the structure of the final material, a gold-decorated carbon microsphere. Calcination temperature not only enhances electrical conductivity but also reduces the active surface area of the material due to the aggregation of gold nanoparticles. Although gold nanoparticles have performed well in increasing the electrocatalytic activity of carbon microsphere, the cost seems to be a barrier for this material to be widely applied. Instead of gold, Ikhsan et al. conducted a study on silver to apply in electrochemical sensors because of their low cost and higher electrical conductivity than noble metals such as gold, palladium or platinum. The rGO–silver nanocomposite was synthesized by modified Tollens' reaction, in which GO with surface functional groups serves as a support material for the growth of silver nanoparticles (Figure 8b) [79]. In the presence of ammonia, glucose is used as a reducing agent. The reduction of $Ag(NO_3)$ (present in the $[Ag(NH_3)_2]^+$ complex) and GO takes place simultaneously, leading to the formation of silver nanoparticles on the reduced GO surface. The presence of silver nanoparticles on the reduced GO surface increases the active surface area, facilitating efficient electron transitions during electrostimulation.

The method of metal oxide immobilization in the structure of carbon materials by oxygen plasma process was proposed by Kim et al. [80] (Figure 8c). First, electrospinning of a homogeneous solution containing polyacrylonitrile (as a carbon source), polystyrene (a pore-forming agent) and $FeCl_3$ (a source of $Fe^{3+}$ ions to create $Fe_2O_3$ nanoparticles) takes place. The oxygen plasma process is performed on electrospun multicore nanofibers and generates multiple oxygen-functional groups on the surface. These functional groups not only increase the cyclization of the polyacrylonitrile chain but also contribute to the phase change of $Fe^{3+}$ ions into $Fe_2O_3$ nanoparticles during stabilization. Finally, carbonization is carried out at a temperature of about 800 °C for 1 min to convert polyacrylonitrile into a carbon structure, with the same progress the polystyrene decomposes and leaves holes on the surface of the final product. For the application in $H_2S$ gas sensors, the large surface area and uniformly dispersed $Fe_2O_3$ component in the carbon structure play a key role in increasing the sensitivity to the target gas (the limit of detection is down to 2 ppm).

Carbon fibers synthesized by electrospinning have emerged as electrode materials because of their high applicability in energy storage devices. However, the carbon structure produced by this method is amorphous carbon with low electrical conductivity, which leads to poor prospects for its further application. To overcome this limitation, Li et al. proposed a new strategy for the synthesis of metal-organic, framework-embedded electrospun nanofibers [81] (Figure 8d). MOF-74(Ni) nanospheres dispersed in electrospun nanofibers, after undergoing pyrolysis and oxidation, are converted into nickel(II) oxide nanoparticles enclosed in carbon nanospheres. This hierarchically mesoporous structure not only increases the space for ion migration but also prevents the elimination of oxides and maintains it in an active state during long-term cycling.

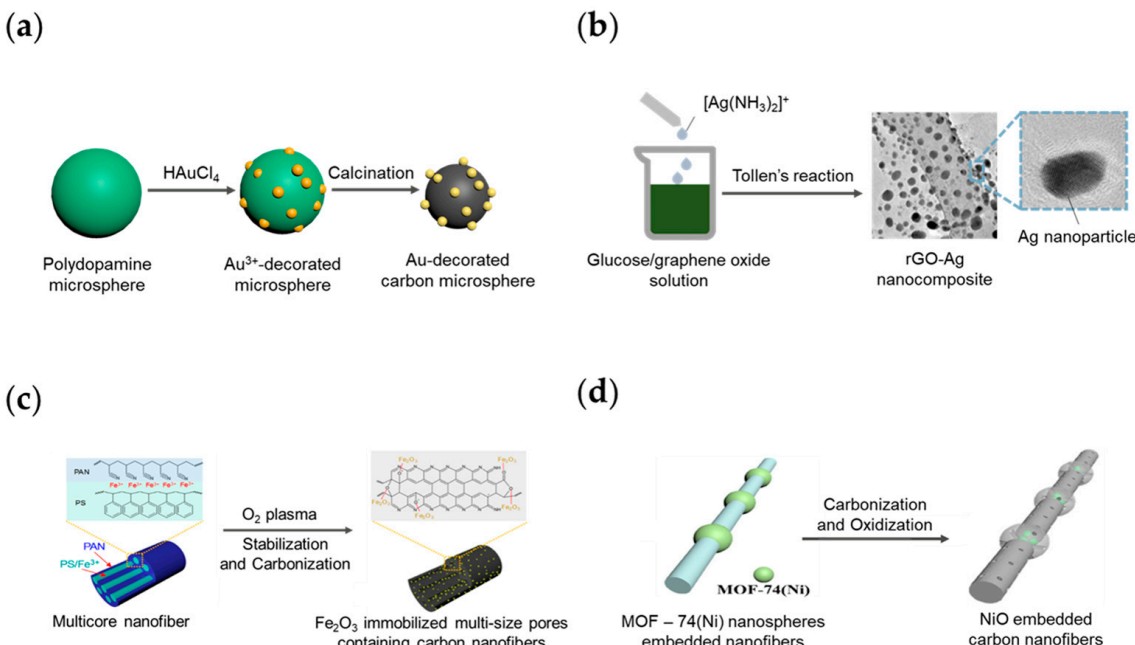

**Figure 8.** (**a**) Carbon spheres surface modified with in situ synthesized gold nanoparticles [78]. (**b**) The rGO-Ag nanocomposite was synthesized by modified Tollens' reaction [79]. (**c**) The plasma oxygen process is used to immobilize iron(III) oxide in the multi-dimensional pore structure of carbon nanofibers [80]. (**d**) MOF-74(Ni) nanospheres dispersed in electrospun nanofibers create mesopores in the structure of the carbon fiber after carbonization and oxidation [81].

## 4. Electro Voltammetry

While electrochemistry deals with the studies based on the relationship between the flow of electrons with the changes in chemicals, the voltammetry electrochemical sensor uses a wild voltage range to determine the existence of a desired target. It uses the oxidase and reduction reactions of the target material, which can be activated when given a suitable amount of energy known as cell potential [82]. The cell potentials of the targets are almost unitary and can be easily distinguished from each other. However, the sensitivity of these reactions greatly depend on the presence of elements of the environment, such as the potential of hydrogen (pH) or the presence of different ions [83]. On the other hand, there are many measurement methods that have been developed throughout history in an effort to achieve high sensitivity and low noise signals. There, the measurement parameters can greatly affect the reaction that happens to the target. Interestingly, the effects of the parameters are unique for each target, so one setting condition may be favorable for one target's response but unfavorable for others. The suitable electrolyte made with simple preparation can help the data generated by voltammetry electrochemical have an outstanding selectivity. On the other hand, when embedding metal/metal oxide nanomaterials into the structure of carbon, the sensing properties will be changed. Since the loaded materials acted as the main catalytic, the position of the peak can be shifted. The loaded material with high electron transfer can keep the redox system reversible even with a high potential change during detecting measurement [84,85]. In addition, the excellent electrocatalytic made the composite more sensitive during the detecting activity, improving the limit of detection. However, it is ineffective for the loaded materials to function alone, since their own structure limit interaction with the electrode needed in the measurement [86].

Currently, there are three most commonly used electrolytes that are applied to the detection of the studies of the materials. The first is simply DI water. Secondly, the detections of the targets can happen ideally in the electrolyte containing $K_3[Fe(CN)_6]$, which acts as an electron transfer mediator due to the highly electrochemical reversibility of

the pair of $Fe(CN)_6^{3-}/Fe(CN)_6^{4-}$ [87,88]. However, not only does this electrolyte need to be prepared in the laboratory and is hard for practical application, but undesirable reactions can also happen between existing ions. Finally, the third is standard phosphate-buffered saline (PBS). PBS, which is now easily available on the market, contains a mixture of ions similar to the physiological one of the human bodies. Hence, PBS is usually used for target detection studies that should be collected from the human body. The downside is that the quality of the peaks generated from the targets is lower than that of the electrolyte containing $Fe(CN)_6^{3-}/Fe(CN)_6^{4-}$.

Nowadays, there are two well-known methods used for the detection of metal ions as well as biomaterials. One is colorimetry, and the other is voltammetry. For the case of colorimetry, although simple preparation steps can be performed at home, it is difficult to obtain information about the target concentration individually. The differences in colors require a complicated system to distinguish them. In the case of voltammetry, while it always requires analytical systems (potential state), the technologies have advanced to the point where many compact/mini-processor systems have been introduced. Voltammetry includes Linear Sweep Voltammetry (LSV), Cyclic Voltammetry (CV), Differential Pulse Voltammetry (DPV) and Square Wave Voltammetry (SWV) (Figure 9). All these measurement methods need to be performed in a three-electrode system. There, the material is usually laid on a commercial carbon electrode, including a glassy carbon electrode (GCE), carbon paste electrode (CPE) and conventional carbon paste electrodes. The recent works based on carbon materials are shown in Table 1.

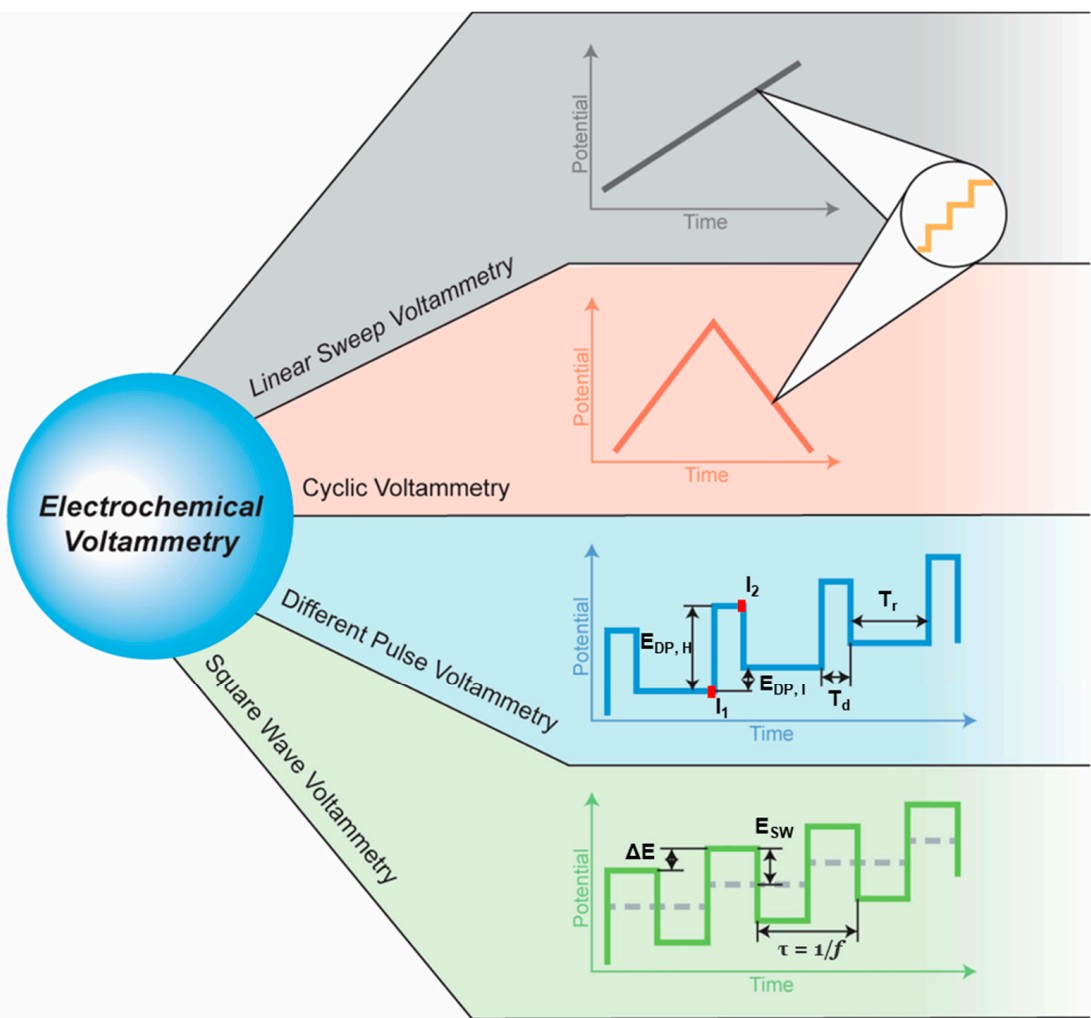

**Figure 9.** Electrochemical voltammetry and measurement methods.

**Table 1.** Carbon material, their variation and performance in detecting application.

| Material Electrode | Measurement Method | Target | Linear Range (μM) | Limit of Detection (μM) | Sensitivity (μA/μM) | Ref |
|---|---|---|---|---|---|---|
| Ag@C/GCE | LSV | Tryptophan | 0.1~100 | 0.04 | 0.05 | [89] |
| RGO–DHP [1]/GCE | LSV | Estradiol | 0.4~20 | 0.077 | nr | [90] |
| $K_3C_{60}$/MPC [2] | LSV | Nitroaromatic compounds | 0.5~240 | 0.17 | 0.303 | [91] |
| AuNPs [3]/GN/GCE | CV | Caffeic acid | 0.5~50 | 0.05 | nr | [92] |
| Activated GCE | CV | Caffeic acid | 0.1~1 | 0.068 | nr | [93] |
| Pd−Cu@$Cu_2O$/N-rGO | DPV | Tryptophan | 0.01~40 | 0.0019 | 0.3923 | [94] |
| Poly(Lmethionine)/GR [4]/GCE | DPV | Tryptophan | 0.05~10 | 0.017 | 0.312 | [95] |
| MIS/MWCNTs-VTMS/GCE [5] | DPV | Caffeic acid | 0.75~40 | 0.22 | 0.39 | [96] |
| rGO/PDA [6] | DPV | Caffeic acid | 0.005~450.5 | 0.0012 | 2.15 | [97] |
| GE/Au/GE/CFE [7] | DPV | Dopamine | 0.59~44 | 0.59 | nr | [98] |
| EBNBHCNPE [8] | DPV | Dopamine | 0.5~160 | 0.2 | 0.1372 | [99] |
| | | Uric acid | 20~600 | 15 | 0.1375 | |
| AuNPs@PDA-rGO | DPV | Riboflavin | 0.02~60 | 0.0096 | nr | [100] |
| | | Pyridoxine | 0.03~600 | 0.025 | nr | |
| CNT/SPCE [9] | DPV | TNT | 0.006~6.6 | 0.006 | 0.44 | [101] |
| PEDOT-Gr/Ta | DPV | Hydroquinone | 5~250 | 0.06 | nr | [102] |
| | | Catechol | 0.4~350 | 0.08 | nr | |
| | | Resorcinol | 6~400 | 0.16 | nr | |
| | | Nitrite | 2~2500 | 7 | nr | |
| $Fe_3O_4$/cMWCNTs/GCE [10] | SWV | Ganciclovir | 0.08~53 | 0.02 | nr | [103] |
| MWCNT/GCE | SWV | Resorcinol | 1.2~190 | 0.49 | nr | [104] |

[1] Dihexadecylphosphate. [2] Macro porous carbon. [3] Gold nanoparticles. [4] Graphene. [5] multiwall carbon nanotubes (MWCNTs)/vinyltrimethoxysilane (VTMS) recovered by a molecularly imprinted siloxane (MIS) film prepared by sol–gel process. [6] Polydopamine. [7] layer-by-layer assembly of graphene sheets and gold nanoparticles modified carbon fiber electrode. [8] 2,2′-[1,2-ethanediylbis(nitriloethylidyne)]-bis-hydroquinone-modified carbon-nanotube-paste-electrode. [9] Screen-printed carbon electrodes. [10] $Fe_3O_4$/carboxylated multi-walled carbon nanotubes modified glassy carbon electrode.

## 4.1. Linear Sweep Voltammetry

LSV or staircase voltammetry, changes the applied potential continuously, usually from the negative to positive potential. When the applied potential becomes favorable for the reaction of the target, the received integration current at the electrode shows peaks that can be clearly distinguished. In fact, the scan speed can directly affect the collected peak width. This phenomenon can greatly limit the ability to identify the chemical in the redox system because there are exit chemicals with peak locations aligned near each other. Although the setup is simple, not only the selectivity provided by the measurement is poor, but the high concentration of materials can also cause noise in the signals. However, its simple mechanism still brings opportunities for application. Masikini et al. conducted a study on estradiol detection (6~20 μM) using another combination of multiwall carbon nanotube (MWCNT) and gold nanoparticles (Figure 10a) [105]. There are many ways to fabricate and combine the excellent properties of nanoparticles and nanotube structures. On the other hand, there are studies proposed that gold nanoparticles and carbon nanotubes can easily support each other and give different advantages for electrochemical properties [106]. Thus, the cyclic voltammetry aera was enhanced and therefore the peak current was higher. However, since the MWCNT and gold nanoparticles were simply mixed together, there is a limitation in the interaction between the two nanomaterials. Filik et al. proposed another MWCNT/gold nanoparticle use for the detection of heavy chromium (Cr) and vanadium (V) ions (Figure 10b) [107]. Here, the authors used neutral red as a cross-linking material between the carbon structure and the gold nanoparticles. This crosslinker allowed electron to be transferred between MWCNT and gold nanoparticles. It is noted that sulfuric acid was used as the supporting electrolyte for the detection. The presence of Cr ions clearly affects the detection of V ions. The nature of LSV made the current value unable to fully recover, making it difficult to see changes in the current

peak. Though the material can detect Cr (0.4~80 μM) and V (3~200 μM) simultaneously, the presence of a high concentration of Cr ions can completely obscure the signal generated by V ions. Punrat et al. introduced the Cr ions detection function of a polyaniline/graphene quantum dot deposited on a screen-printed carbon electrode using cyclic voltammetry (Figure 10c) [108]. Conducting polyaniline is capable of improving the sensitivity of electrochemical detection thanks to its stable electrical conductivity [109]. The authors emphasize the advantage of LSV, which is the short operation time to create a rapid measurement device. The sensor appears to be able to detect the presence of Cr (0.1~10 ppm) in a short amount of time, repeatedly, over multiple times without the need for renewal. However, the composite has the morphology that is similar to a film, which may have limited its surface area. Li et al. developed a nitrogen-doped carbons by treating graphene oxide precursor with melamine (Figure 10d) [110]. The material was applied in the application of ascorbic acid (0.6~1.2 mM), dopamine (0.12~0.22 mM) and uric acid (0.1~0.25 mM) detection. The material can detect the presence of three chemicals independently in the same test. Thus, the results are a method for the development of a rapid and good material for the detection of the above biomaterials.

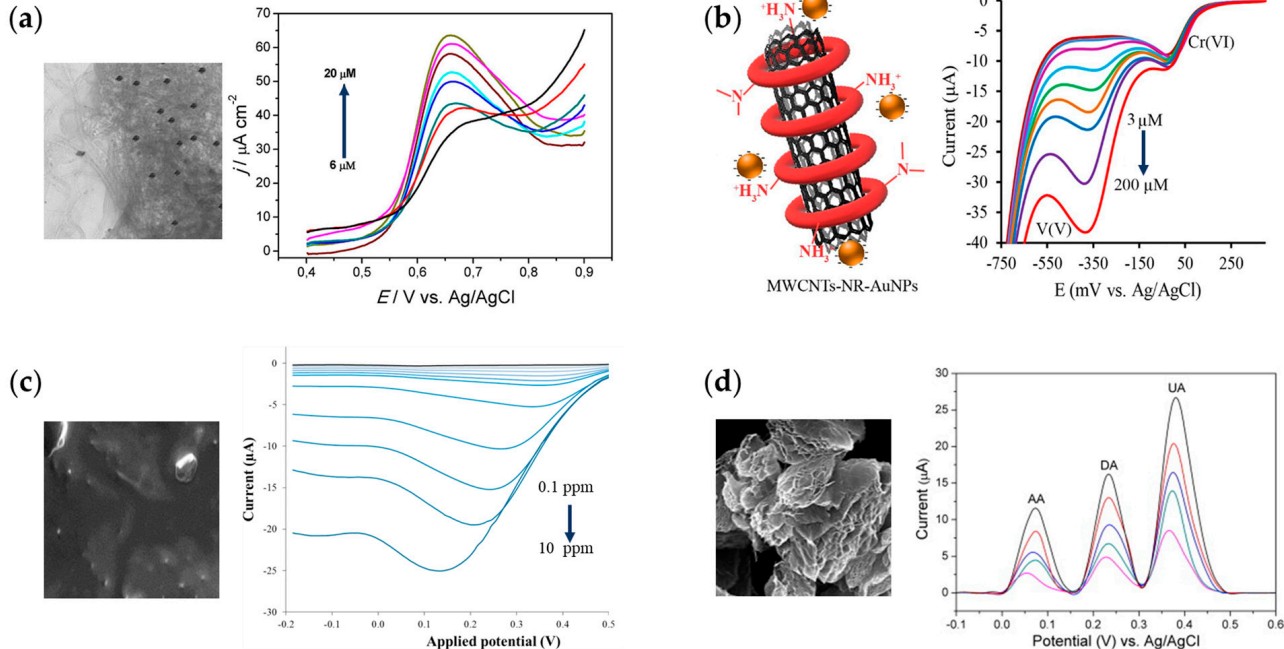

**Figure 10.** (**a**) Multiwall carbon nanotube/gold nanoparticle mixer for estradiol detection [105]. (**b**) Gold nanoparticles decorated on multiwall carbon nanotube for the detection of Cr and V ions [107]. (**c**) Polyaniline/graphene quantum dot on the screen-printed carbon electrode toward sensitivity of Cr ions [108]. (**d**) Nitrogen–doped graphene for electrochemical detection of ascorbic acid, dopamine and uric acid [110].

*4.2. Cyclic Voltammetry*

CV has the same mechanism as linear sweep voltammetry, but the scanning of potential is made into a closed cycle. After the oxidation process has been carried out, the reduction reaction will occur when the second scanning process occurs. The closed cycles of the scanning potential allow the material to be recovered to its original state, and from there it can be reused. On the other side, the repeatability of cyclic voltammetry ensures that the oxidation state of the material is the same for each measurement. Thus, this measurement method has high repeatability while requiring a short measurement time. Karikalan et al. conducted research on the caffeic acid sensor with nitrogen-doped carbon (Figure 11a) [111]. The authors used a method in which pyrrole was burned to prepare the carbonaceous materials with a hetero atom dopant. The current peak linear range of the sensor is from

100 to 1000 μM. It is noted that the material is also able to sense the presence of caffeic acid by DPV with an ultra-low limit of detection at 0.01 μM. Karthika et al. used graphite carbon nitride as a material decorated with AgM (Figure 11b) [112]. Graphite carbon nitride has excellent redox properties, is environmentally friendly, has good stability and has high electrical conductivity. The composite material was used to detect heavy Cr ions in a pH 2 environment. The studied carbon material can show the signal in the CV curve with Cr ions from 10 to 100 μM present in the solution. Zhang et al. worked on graphite carbon nanosheets based on laser-scribed graphenization from a polyimide precursor (Figure 11c) [113]. It was mentioned that the preparation process of the material was in a matter of minutes. After the carbon material is prepared, the palladium was decorated by the electrodeposition method. Finally, the composite material can detect hydrazine (50 μM~5 mM), which is harmful to the human body despite its wild applications. However, it is clear that the limit of detection was sacrificed for production efficiency.

The combination of MWCNT and a manganese-based metal-organic framework was introduced by Madej et al. (Figure 11d) [114]. The authors applied the material for the detection of citalopram (0.05~115 μM). Although the anodic oxidation peak results collected from the CV cannot be calibrated to one linear line, they form a linear line in three concentration ranges with three different sensitivities. The research showed that using the high electrocatalytic material as the loaded material in the composite can greatly improve the sensing function in voltammetry.

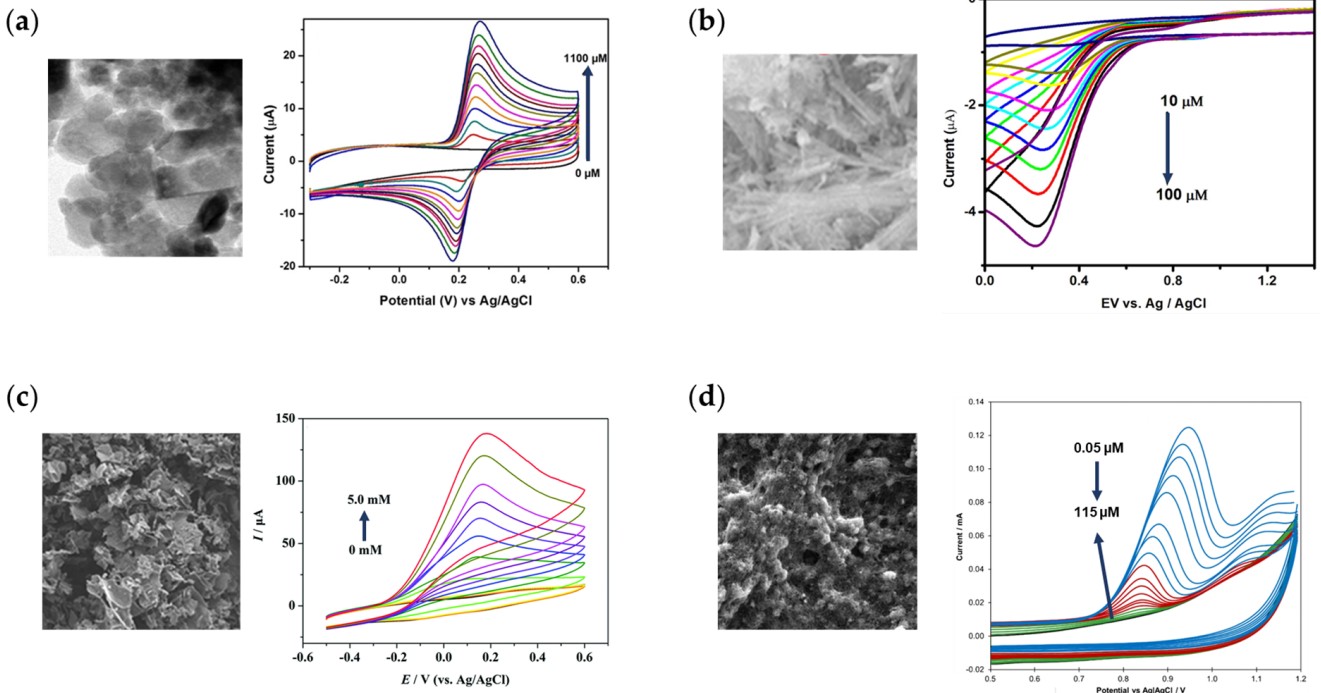

**Figure 11.** (**a**) Voltammetry determination of caffeic acid in red wines using nitrogen-doped carbon material [111]. (**b**) Graphene carbon nitride-doped silver molybdate immobilized Nafion for sensitivity of Cr ions [112]. (**c**) Ultra sensitivity toward hydrazine based on palladium−loaded laser−scribed graphite carbon nanosheets [113]. (**d**) Carbon/metal–organic framework composite for citalopram detection [114].

## 4.3. Different Pulse Voltammetry

Since the reactions surrounding the electrode are diffusion processes, the scanning rate may directly affect the accuracy of the data received from the electrochemical system [115]. The slower the scan rate, the better the peak current. Despite the advantages of the potential state, it is difficult to make such an accurate small change in the applied potential. Thus,

the DPV, which involves the use of a potential waveform, was developed to overcome the problem. Instead of simply using the measured current, the change in current during the execution of the pulse was calculated (I1 and I2) (Figure 9), where Tr is the rest time, Td is the pulse duration, EDP, H is the pulse high and EDP, I is the pulse increment. In exchange for the low limit and wild range of detection, there are many parameters that need to be controlled.

Sakthivel et al. introduced a combination of VS2-SNS2 and functionalized MWC-NTs that can recognize the presence of the neurotransmitter dopamine (Figure 12a) [116]. Transition metal sulfides have excellent physicochemical properties that are suitable for electrochemical sensors [117,118]. The detection limit could be reduced to 8 nM, and the linear range was extended to 25~1017 μM. This is the typical behavior of the DVP, which has excellent sensitivity yet a long measurement time. Motaghedifard et al. used sulfated zirconium oxide mixed with polyaniline nanostructure to sense Cr ions in wastewater (Figure 12b) [119]. Here, instead of using carbon as the conducting material, the authors used polyaniline to transfer charge between zirconium oxide and the measurement system. The material again showed high sensitivity with a limit of detection of 64 nM, and a linear range line between 0.55 and 39.5 μM. The application of anionic surfactant sodium lauryl sulfate modified carbon nanotube and a pencil graphite composite paste electrode for the detection of riboflavin was introduced by Tigari et al. (Figure 12c) [120]. Their sensor material can detect a target within the range of 0.2 to 5 μM. Although the upper point of the linear range is low, the material showed an excellent limit of detection of 12 nM. Manasa et al. proposed MWCNT embedded with maghemite nanoparticles (Figure 12d) [121]. The material was used to detect resorcinol, which is a concern for ecology and human health. The sensor also reported a limit of detection of 22 nM, and a linear range from 0.5 to 10 μM. Though the peak current calibration in the mentioned studies was divided into two parts, they still form linear lines. Compared to the staircase voltammetry, the DPV measurement technique is clearly preferable for better results at the limit of detection and can almost reduce the noise generated by redox reactions.

*4.4. Square Wave Voltammetry*

Since the execution of DPV takes a long time, SWV was proposed to optimize the time consumption of the measurement process. The basic mechanism of SWV is the same as DPV but the rest time is equal to the duration of the pulse, forming the waveform of the applied potential over time. Reducing the rest time can make a big effect on the shape of the current peak. The peak current will increase, the peak width will decrease, and the peak location will shift to a nearby location [122]. The parameters included in SWV are pulse increment ($\Delta E$), square wave amplitude (ESW) and duration of the potential step $\tau$ ($1/f$, where f is the frequency of the wave) (Figure 9) [123].

Castro et al. studied the sensitivity of reduced graphene oxide/MWCNT nanocomposite toward 2,4,6-trinitrotoluene (Figure 13a) [124]. The combination of the two carbon materials seems to have a huge impact on the detection ability, including the limit of detection, linear range, and limit of quantification. The limit of detection is 60 nM, which is much lower than for the two materials separately. The linear range also becomes wilder, extended to 0.5~1100 μM. This advantage and the low interfacial resistance between the material electrode and the electrolyte may be due to the synergistic effect. A composite poly(amidoamine) dendrimer functionalized magnetic graphene oxide prepared by co-precipitation was introduced by Baghayeri et al. (Figure 13b) [125]. Graphene oxide was bound to poly(amidoamine) through the reaction between the amine group of the polymer ($-NH_2$) and the acid functional group of graphene ($-COOH$). Not only can the material detect palladium (Pd) and cadmium (Cd) ions, but it can also distinguish the concentration of ions individually. While the linear range for the two ions is said to be the same (0.5~100 μM), the sensitivity of the Cd ions is lower. Phan et al. used the modified carbon nanoparticles that were carbonized from a polypyrrole precursor (Figure 13c) [126]. The authors used the plasma field in the exposure with $O_2$, $NH_3$ and $C_4F_8$ in order to dope

different elements in the structure of amorphous carbon material. The electrochemical properties of carbon treated with $O_2$ plasma seemed to be the most enhanced, followed by $NH_3$ plasma and $C_4H_8$ plasma. Since the material is formed from polypyrrole, there are many nitrogen-functional groups already doped in the structure of carbon. $NH_3$ plasma only increases the number of nitrogen function groups and $O_2$ plasma brings out the combination of oxygen and nitrogen elements. The developed oxygen-doped carbon nanoparticles have the limit of detections 5 and 10 nM toward lead (Pb) and copper (Cu) ions, respectively. Though it is well known that doping functional group can enhance the sensing activity of material, this study proved that oxygen-doping and nitrogen-doping are the most effective. Nevertheless, plasma seems to be a very useful method for element-doping, since it is fast, safe to use and environmentally friendly. Brycht et al. studied the difference in fenhexamid sensitivity of basic carbon electrodes such as glassy carbon electrodes, glassy carbon paste, conventional carbon paste electrodes and carbon paste electrodes modified with MWCNTs (Figure 13d) [127]. When comparing basic electrodes, the conventional carbon electrode has advantages in both the linear range and limit of detection. On the other hand, though the modification of the electrode with MECNTs clearly increases the linear range and limit of detection, it may limit the recovery of the electrode between measurements.

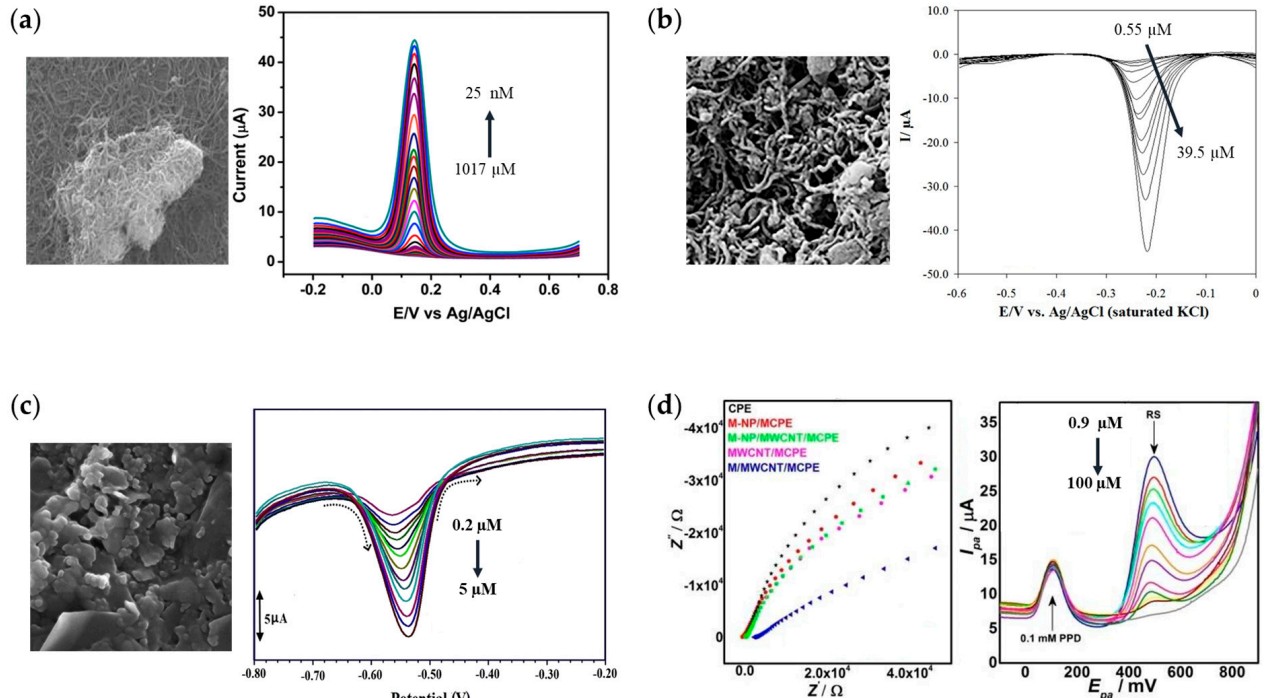

**Figure 12.** (**a**) Determination of the neurotransmitter dopamine by DPV on binary metal sulfides hybrid/functionalized MWCNTs [116]. (**b**) Sensitivity toward Cr in wastewater via polyaniline/sulfate zirconium dioxide/MWCNTs [119]. (**c**) Composite of pencil graphite and carbon nanotubes and its sensitivity to riboflavin [120]. (**d**) Rapid quantification of resorcinol in hair dye using Maghemite/MWCNTs composite [121].

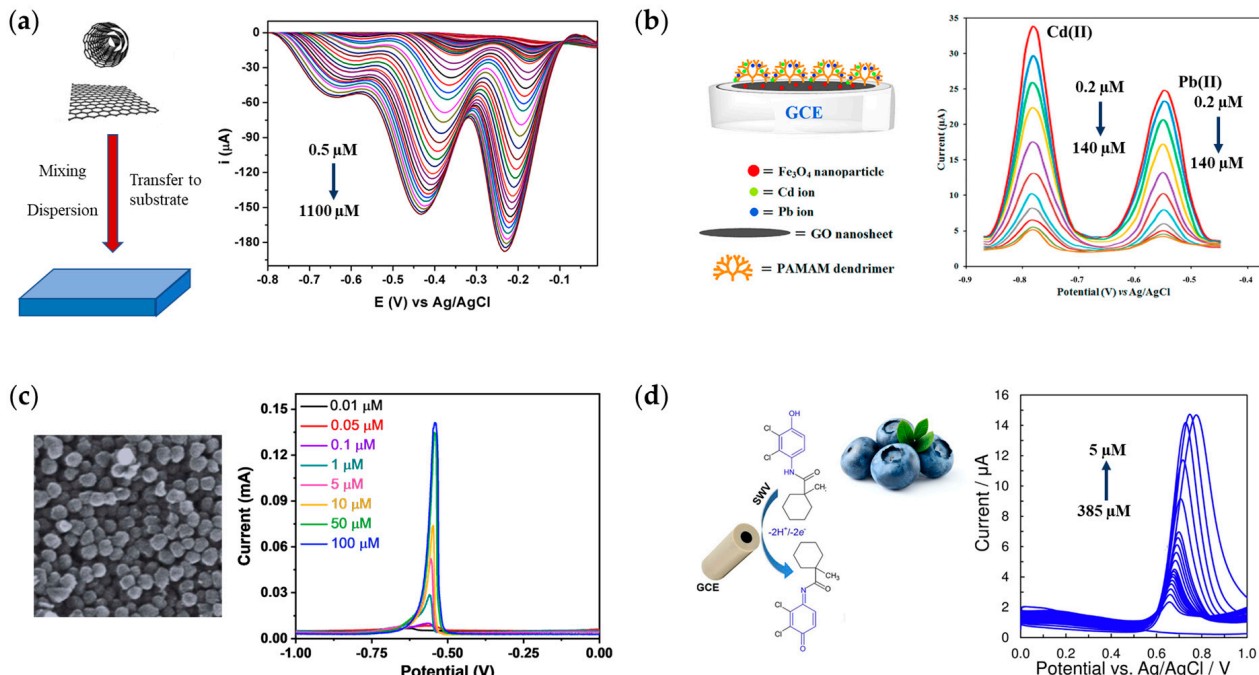

**Figure 13.** (**a**) The nanocomposite of reduced graphene oxide/MWCNTs for high sensitivity of trinitrotoluene) [124]. (**b**) Functionalized magnetic graphene oxide for the determination of Pb and Cd [125]. (**c**) Carbon nanoparticles modified with plasma towards ultrasensitivity of heavy metal ions [126]. (**d**) Fungicide fenhexamid in berries and wine grapes determination using carbon−based electrode [127].

## 5. Conclusions

This review presents carbon nanomaterials and their application in electrochemical voltammetry. $sp^2$ carbon nanomaterials can come from a variety of sources. Depending on the properties of the carbon material, it may have an amorphous or crystallizing structure. To date, although many carbon nanomaterials have been developed, the multiwalled carbon nanotube seems to be the most widely used. The detection process uses voltammetry methods, including linear sweep voltammetry, cyclic voltammetry, different pulse voltammetry and square wave voltammetry. Until recently, more and more new carbon materials have been developed for better properties and morphological structure. On the other hand, the voltammetry methods still have great potential for the determination of many other targets.

**Author Contributions:** Writing—original draft preparation, T.D.N. and M.T.N.N.; review and editing, T.D.N., M.T.N.N. and J.S.L.; supervision, J.S.L.; project administration, J.S.L. All authors have read and agreed to the published version of the manuscript.

**Funding:** This research received no external funding.

**Data Availability Statement:** Not applicable.

**Acknowledgments:** This work was supported by the National Research Foundation of Korea (NRF) grant funded by the Korea government (MSIT) (No. NRF-2021R1F1A1053291) and Gachon University Research Fund of 2021 (GCU-202106560001).

**Conflicts of Interest:** The authors declare no competing financial interest.

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
