# Peer review of "Carbon-Based Materials and Their Applications in Sensing by Electrochemical Voltammetry"

_inorganics, doi:10.3390/inorganics11020081_

Round 1
Reviewer 1 Report
This is a well-structured review on carbon-based materials for electrochemical sensing. The authors summarized the previously-reported literatures with their synthesis methods, electrochemical performance, and design highlights. However, there are some remaining questions on the active sensing center. Thus, I believe a minor revision is needed for this manuscript at this stage.
Specific Comments:
1. Some of the graphene-like carbon materials reported to have detection functions are benefiting from the loaded materials instead of themselves. How to distinguish the intrinsic sensing properties vs. the add-on properties from other materials? Also, the carbon materials only serve as conductive substrates should be noted separately.
Author Response
COMMENT
This is a well-structured review on carbon-based materials for electrochemical sensing. The authors summarized the previously-reported literatures with their synthesis methods, electrochemical performance, and design highlights. However, there are some remaining questions on the active sensing center. Thus, I believe a minor revision is needed for this manuscript at this stage.
RESPONSE:
We highly appreciate your positive comments on our manuscript.
SPECIFIC COMMENTS:
1) Some of the graphene-like carbon materials reported to have detection functions are benefiting from the loaded materials instead of themselves. How to distinguish the intrinsic sensing properties vs. the add-on properties from other materials? Also, the carbon materials only serve as conductive substrates should be noted separately.
RESPONSE:
Thank you for your comment about distinguish of the pristine sp2 carbon and carbon compoistes in electrochemical sensing application. As you mentioned, the loading of additional metal/metal oxide material can contribute directly to the detecting function of the carbon material. These materials have excellent electrocatalyst properties. Depending on the individual, many changes may happen to the sensing properties:
1) The overpotential may be changed leading to the shift of the peak potential.[R1]
2) The sensitivity of the material can be significantly improved, and the limit of detection can go down lower.
3) The highly active surface area improves the interaction of composite material with the detecting target.
- Faster charge transfer benefits from conducting loaded material.[R2]
On the other hand, graphite-like carbon materials have been developed in many studies to achieve high surface area and pore size by modifying the morphology of the carbon material. Thus, the material has higher sensitivity, larger linear range, and lower limit of detection. The chemical structure of carbon material has also been modified with a variety of methods to doping functional groups carrying atoms like oxygen, nitrogen, or fluoride. The presence of these functional groups can greatly enhance the interaction between carbon material with electrolyte, then reduce surface resistance.[R3]
Hence, there are similarities in the improvement in sensing properties of the carbon and carbon-metal/metal oxide composite materials. However, the loaded material may shift the location of the peak signal to a considerable amount compared to intrinsic carbon material.[R1,R4,R5] The composite materials also can be operated at a higher scan rate than intrinsic carbon while keeping the redox system reversible.[R6]
As you requested, “metal/metal oxide composites” has been mentioned separately as follows in section 3 and 4.
The following paragraphs were added in the section 3 of the revised manuscript.
(1) “Unlike the material that was mainly built based on the carbon structure, the carbon composite was designed so that the metal/metal oxide has the largest active surface area. The composite’s detecting function will be benefiting from the loaded material. Here, sp2 carbon plays the role of the bone structure, which conducts the electricity between the electrode and the loaded materials. Instead, metal/metal oxide material will exchange electrons with substances in the electrolyte. Though metal oxide material does not own high electrical conductivity like metal materials, they both have excellent electrocatalyst ability, resulting in higher sensitivity.[R6, R7] It is noted that some of the metal oxides like SnO2 and ZnO can carry the properties of a semiconductor material, which made them good materials in electron transfer ability.[R8, R9] Fe3O4 and Co3O4 with multiple valence states have also been found to be outstanding in their performance in catalytic activities.[R10, R11]
A method for dispersing gold nanoparticles in carbon microspheres, that is non-toxic, has been proposed by Zhou et al (Figure R1a).[R2] The Au3+-decorated microspheres were obtained by simply dissolving HAuCl4 in the suspension of polydopamine. Here, polydopamine acts not only as an adsorbent, reducing agent, and stabilizer but also as a carbon source for the structure of the final material, a gold-decorated carbon microsphere. Calcination temperature not only enhances electrical conductivity but also reduces the active surface area of the material due to the aggregation of gold nanoparticles. Although gold nanoparticles have performed well in increasing the electrocatalytic activity of carbon micro-sphere, the cost seems to be a barrier for this material to be widely applied. Instead of gold, Ikhsan et al. conducted a study on the silver to apply in electrochemical sensors because of their low cost and higher electrical conductivity than noble metals such as gold, palladium, or platinum. The rGO-silver nanocomposite was synthesized by modified Tollen's reaction, in which GO with surface functional groups serves as a support material for the growth of silver nanoparticles (Figure R1b).[R1] In the presence of ammonia, glucose is used as a reducing agent. The reduction of Ag(NO3) (present in the [Ag(NH3)2]+ complex) and GO takes place simultaneously, leading to the formation of silver nanoparticles on the reduced GO surface. The presence of silver nanoparticles on the reduced GO surface increases the active surface area facilitating efficient electron transitions during electrostimulation.”.
Figure R1. (a) Carbon spheres surface modified with in situ synthesized gold nanoparticles [R2]. (b) The rGO-Ag nanocomposite was synthesized by modified Tollen's reaction [R1].
The following paragraph was added in the section 4 of the revised manuscript.
(2) “On the other hand, when embedding metal/metal oxide nanomaterials into the structure of carbon, the sensing properties will be changed. Since the loaded materials acted as the main catalytic, the position of the peak can be shifted. The loaded material with high electron transfer can keep the redox system reversible even with a high potential change during detecting measurement.[R12, R13] And the excellent electrocatalytic made the composite more sensitive during the detecting activity, improving the limit of detection. However, it is ineffective for the loaded materials to function alone, since their own structure limit interaction with the electrode needed in the measurement.[R5]”.
[R1] Chemicalchim. Acta, 2016, 192, 392-399.
[R2] Sens. Actuators B Chem., 2016, 237, 487-494.
[R3] Synth. Met., 2022, 291, 117203.
[R4] Chemicalchim. Acta, 2013, 106, 127-134.
[R5] J. Hazard. Mater., 2017, 333, 54-62.
[R6] Chem. Soc. Rev., 2019, 48, 2518-2534.
[R7] Microchim. Acta, 2016, 222, 717-727.
[R8] Mater. Sci. Eng. C, 2017, 71, 386-394.
[R9] Microchim. Acta, 2010, 55, 2835-2840.
[R10] Sens. Actuators B Chem., 2019, 281, 1063-1072.
[R11] Analyst, 2012, 137, 2840-2845.
[R12] Chemtexts, 2016, 8.
[R13] J. Chem. Educ., 2018, 95(2), 197-206.
Revised parts in the manuscript
(1) Contents related to the carbon composite material are described in more detail independently of section 3. (see Line 260, Page 9)
(2) Paragraph (2) was added to section 4 in the revised manuscript. (see Line 263, Page 9)
(3) Figures R1 were added to the revised manuscript as Figures 8. (see Line 318, Page 10)
(4) Reference [R1] was added to the manuscript as [79].
(5) Reference [R2] was added to the manuscript as [78].
(6) Reference [R5] was added to the manuscript as [86].
(7) References [R6-R11] were added to the manuscript as [72-77].
(8) References [R12, R13] were added to the manuscript as [84, 85].

Reviewer 2 Report
The authors have written a well-balanced and up to date literature review. It reads easily and is meant to be an incremental progress report rather than a full review. A major problem with this kind of reviews is their current saturation in literature. However, this manuscript covers sections that have not been oft-reviewed. Much of the research reviewed herein are very recent examples in the field from the past few years.
Overall it is a good publication. My major reservation on this paper were simply the lack of a critical review between the studies. I realise that this is not a given for simple literature reviews. However, without that, I read this study as another small incremental review that is too general to be of particular use to the member of this field.
In summary, while much of this review is excellent, more useful would be the use of the platform created by the authors to critically address specifically where the gaps in this field currently lie.
Author Response
COMMENT
The authors have written a well-balanced and up to date literature review. It reads easily and is meant to be an incremental progress report rather than a full review. A major problem with this kind of reviews is their current saturation in literature. However, this manuscript covers sections that have not been oft-reviewed. Much of the research reviewed herein are very recent examples in the field from the past few years.
Overall it is a good publication. My major reservation on this paper were simply the lack of a critical review between the studies. I realise that this is not a given for simple literature reviews. However, without that, I read this study as another small incremental review that is too general to be of particular use to the member of this field.
In summary, while much of this review is excellent, more useful would be the use of the platform created by the authors to critically address specifically where the gaps in this field currently lie.
RESPONSE:
Thank you for your positive comments about this paper. As you mentioned, more critical reviews are added in the revised manuscript.
Revised parts in the manuscript
(1) The sentence “Because GO has good solubility in water/organic solvents and has a favorable electron mobility structure, it becomes an attractive material for electrochemical studies and applications.” was added in the revised manuscript. (see Line 87, Page 3)
(2) The sentence “Moreover, this study is the premise for several future studies related to the hydrothermal synthesis of rGO, which can be applied in enhancing photocatalytic and improving the electrochemical performance of electrodes.” was added in the revised manuscript. (see Line 139, Page 4)
(3) The sentences “Here, electrospinning method is used to synthesize 1D structure (fiber). This method allows control over the structure of the material by varying the composition of the precursor. For instance, polyacrylonitrile can undergo carbonization in the temperature range of 800 – 1300 °C, and poly methyl methacrylate will be degraded at about 300 °C. Therefore, poly acrylonitrile act as the carbon source of material, and the decomposition of poly methyl methacrylate create the pore structure for the carbon nanofibers.” were added in the revised manuscript. (see Line 160, Page 5)
(4) The sentences “The discovery of the carbon material containing a fully sp3 amorphous structure with outstanding physical properties added a diamond-like structure to the carbon family. Such material seems to have a high potential for further exploitation.” was added in the revised manuscript. (see Line 178, Page 6)
(5) The sentence “However, since the MWCNT and gold nanoparticles were simply mixed together, there is a limitation in the interaction between the two nanomaterials.” was added in the revised manuscript. (see Line 399, Page 14)
(6) The sentence “This crosslinker allowed electron to be transferred between MWCNT and gold nanoparticles” was added in the revised manuscript. (see Line 404, Page 14)
(7) The sentence “However, the composite has the morphology that is similar to a film, which may have limited its surface area” was added in the revised manuscript. (see Line 417, Page 14)
(8) The sentence “However, it is clear that the limit of detection was sacrificed for production efficiency.” was added in the revised manuscript. (see Line 454, Page 15)
(9) The sentence “The research showed that using the high electrocatalytic material as the loaded material in the composite can greatly improve the sensing function in voltammetry.” was added in the revised manuscript. (see Line 460, Page 15)
(10) The sentence “Here, instead of using carbon as the conducting material, the authors used polyaniline to transfer charge between zirconium oxide and the measurement system. was added in the revised manuscript.” (see Line 487, Page 16)
(11) The sentence “Though it is well known that doping functional group can enhance the sensing activity of material, this study proved that Oxygen-doping and Nitrogen-doping are the most effective. Besides, plasma seems to be a very useful method for element-doping, since it is fast, safe to use, and environmentally friendly.” was added in the revised manuscript. (see Line 542, Page 18)
